# A Novel Development Scheme of Mobility as a Service: Can It Provide a Sustainable Environment for China?

**Zipeng Zhang** * **and Ning Zhang**

School of Economics and Management, Beihang University, Beijing 100191, China; ning_zhang@buaa.edu.cn
* Correspondence: zipengzhang@buaa.edu.cn

**Abstract:** Mobility as a service (MaaS), a new concept of transportation, is regarded as an effective solution to prevalent urban traffic problems because of its sustainable development properties such as sharing, integration, humanism and low-carbon. However, progress from pilots to large-scale implementation has hitherto been slow especially in China. In this paper, we propose a new alliance-based framework of development scenarios for the Chinese MaaS system. At the same time, by distinguishing the experience and lessons between the UbiGo project in Sweden and Whim project in Finland, we summarize that the key to the success of prior pilot projects is the cooperation of industry alliance, government policy support, and data sharing mechanism. Moreover, this paper also proposes some MaaS solutions for the obstacles of alliance-based cooperation, data resource sharing, business model selection, operation conditions, development path, policy support and other aspects in China.

**Keywords:** MaaS; development scenarios; comparative analysis; pilot experience

## 1. Introduction

The expansive and densely populated cities provide convenient and diverse mobility services to modern lives, and meanwhile, they also bring a series of negative impacts to the urban transportation system: increasingly severe traffic congestion [1,2], air pollution [3–5], and unsustainable transportation and mobility problems caused by imbalances in supply and demand [6,7], which become the main dilemma for city planners and managers. The lack of integration and inconsistency in the connection process between transportation modes (such as the last-mile problem) not only reduces the operational efficiency of urban transportation and public transportation systems but also reduces the reliability of multi-modal transportation. In recent years, the rise in the number of cars has also caused serious livelihood problems such as traffic congestion, traffic pollution, and difficulty in parking, which has reduced urban traffic satisfaction. Otherwise, the explosive growth of the shared mode of transportation and its subsequent problems has caused extensive discussion on the re-examination and re-thinking of current mobility service modes such as the problem of conflicts between taxis and app-based car-sharing service [8,9] and the problem of re-congestion caused by excessive empty driving of ride-sharing service cars [10]. It is generally believed that the mobility oriented development (MOD) that blindly increases the supply of a single mode of transportation and ignores the coordination and cooperation between multi-mode mobility systems cannot meet the requirements of being efficient, comfortable and sustainable in future urban development [11–13]. We urgently need a novel mobility choice to meet the needs of the changing concept of travel in the process of urban traffic development to reduce the various problems faced by urban transport travel. Mobility as a service (MaaS), a new transportation travel concept containing four main features: sharing, integration, humanism, and low-carbon, develops from the integration of travel information, sharing mobility modes and mobile information technology. This feature provides ideas to meet the increasing travel demand and in the meanwhile gradually

replace private driving for solving the current traffic dilemma and travel problems in city areas.

The preceding literature review raises important questions on the impacts of emerging MaaS patterns on travel behavior, vehicle ownership and traffic efficiency. Specifically, the advent of interactive and collaborative MaaS is leading a push towards smaller and more flexible (point-to-point) modes of transport. Questions remain surrounding the sustainability of various business development scenarios, and their impacts on the existing modes of transport services. What are the implications of the MaaS pattern making future technologies such as autonomous and connected vehicle technologies come online? How will current policies of public transport subsidy and private car ownership be affected? What are the whole-scale impacts of the MaaS system on road capacity, traffic congestion, land use and the urban layout? Apparently, while the MaaS paradigm is still in the nascent stage, it is difficult to answer all of these questions in just a dozen pages, it is necessary to consider the plethora of new transport modes (including distinct ownership models) enabled by various digital disruptors and develop a framework with which to evaluate their suitability for a variety of urban environments. Accordingly, this paper explores alliance-based development scenarios for MaaS and a framework for the future development of MaaS in China. More precisely, the paper examines the following, from the perspectives of incumbent private transport participators and third-party participators that are interested in roles in the nascent MaaS system and give answers to the questions below:

- How to ensure that the city has a good implementing foundation of the MaaS ecosystem in China?
- Who should be the provider, integrator and operator of the MaaS ecosystem in China?
- Which business development paradigm (BDP) is better for Chinese MaaS foundations?
- How to learn from success and failure in the current pilot?
- How to achieve technology integration in China, such as driverless cars, block chain and artificial intelligence?
- How to advance data sharing for achieving the value of mobility big data in China?

The rest of the paper is organized as follows. Section 2 presents the review of the literature. In Section 3, we describe the key properties and successful foundations of the MaaS system. In Section 4, we discuss the features of the Kamargianni model and Smith model in public/private ownership development paths of MaaS, and this paper develops a new alliance-based development scenario model in this section. This paper also introduces the pilot experience in Sweden and Finland in Section 5, which are verified in the case of China in Section 6.

## 2. Review of the Literature

### 2.1. Theoretical Research

The MaaS concept, first proposed by Hietanen [14], defined and developed by Karlsson et al. [15], is an emerging research area that has attracted widespread attention from scholars and transportation researchers in recent years. Most researchers believe that the rapid development of shared travel modes in cities, the change in car ownership attitudes, and the normalization of urban traffic congestion, have promoted the formation of the MaaS integration concept.

Although there have been some overview articles [16–19] in recent years, MaaS is still in a nascent state of concept considering the number of structurally analyzed pilots is small. As a consequence, empirical observations such as thought pieces and technical reports flood the MaaS research literature, mathematical analyses and numerical simulations for the MaaS diffusion model are by and large yet to be conducted, which still arguably has been an integral contributor to the development of MaaS. Notable streams within the articles, either presented as graduation dissertations or published in peer-reviewed academic conferences and journals include (1) frameworks and features of MaaS (e.g., [16,20]); (2) considerations of viable development scenarios for MaaS based on current conditions and needed policies (e.g., [21–28]); (3) studies on feasible profit models such as the bundling and pricing scheme

for MaaS (e.g., [29–31]); (4) Assessments of potential users, such as preference and behavior (e.g., [32–35]); (5) assessments of effects in multi-mode interaction of MaaS diffusion such as community transport and public transit (e.g., [36–38]); (6) Explorations of the pilot all over the world (e.g., [22,25,28,37–42].

In frameworks and features of MaaS aspects, [43] summarizes and defines nine core characteristics for the study of MaaS, which provide a theoretical basis for the research of the potential impact of travel behavior and travel preferences in the MaaS system, Sochor et al. [44] proposes a widely accepted MaaS classification framework (as shown in Table 1), this framework divides MaaS into four development stages based on the degree of mobility modes integration, information integration and the type of travel services provided. In terms of viable development scenarios, the MaaS system mainly faces the balance issue of contradiction between emerging business progress and lagging government regulation (permission in policy and data). In order to integrate all potential mobility resources, Kamargianni et al. [45] introduces a new player, which they refer to as the MaaS provider. The MaaS provider purchases mobility assets, capabilities or services from other transport service providers, then integrates them with public transport services and sells collection packages of multi-modal mobility service to end-users. According to the different roles of government regulation and the open degree of sharing data, Smith et al. [21] divided MaaS into three different development scenarios by introducing two new MaaS roles (MaaS integrator and MaaS operator). These three scenarios, the market-driven development model (opened data), public control development model (Private data) and public-private mixed development model (partially opened data), provides a reference for the development of MaaS in different travel market backgrounds. As a provider of traditional urban public transportation, the positioning and roles of government departments in the emerging MaaS system have also been widely discussed. Kamargianni et al. [46] advocated that the MaaS ecosystem should be held by the government, whose public power not only helps easily integrate other mobility services into public transport services but also provides easier access to user support. At the same time, privacy and information protection issues about travel and privacy information produced in the using process of MaaS have also become a key direction of continuous optimization and exploration of the MaaS system [34].

**Table 1.** Characteristics of the Development Stages of MaaS system.

| Integrated Content | Travel Modes and Services | MaaS 1.0 | MaaS 2.0 | MaaS 3.0 | MaaS 4.0 |
|---|---|---|---|---|---|
| **Travel Information** | Interactive/multi-mode | ✓ | ✓ | ✓ | ✓ |
| | Travel planning and price estimation | ✓ | ✓ | ✓ | ✓ |
| **Booking and Payment** | Reservation and payment (single trip) | ✗ | ✓ | ✓ | ✓ |
| **Mobility Services** | Bundle subscription and long-term contract | ✗ | ✗ | ✓ | ✓ |
| **Social Goals** | Traffic congestion and urban layout optimization | ✗ | ✗ | ✗ | ✓ |

### 2.2. Pilot Research

The MaaS concept aroused the widespread concern of the participants in the ITS (Intelligent Transport System) European Conference organized in Helsinki, Finland in 2014. In 2015, the world's first regional MaaS alliance, the European MaaS Alliance, was established in France. The main members of the alliance include transportation service providers, public transport operators, MaaS operators, integrators, IT system providers, users and governments at all levels. Since then, MaaS has rapidly gone from nowhere to nearly everywhere in the personal mobility service sector. After that, a MaaS software app

"Whim", which provides multi-modal transportation services for cities around the world, was launched and demonstrated in Helsinki, Finland. It provides a series of multi-modal travel packages and pay-as-you-go plans to allow city commuters to book and pay for buses, trains, taxis, bicycles, car rental and other travel services just using one convenient software through a smart mobile phone. Since then, numerous MaaS pilot programs and applications have been performed, including NaviGoGo in Scotland, the United Kingdom; SMILE in Vienna, Austria; IDPASS in France; Maishi Mobility in Shenzhen, China (for more details, please see Table 2). In September 2019, the Chinese government issued the Outline for the Construction of a Powerful Transportation Country, which clearly stated: "Strengthen the development of shared transportation, build a service system based on mobile smart terminal technology, and realize mobility as a service".

**Table 2.** Overview of Popular Pilot APP of MaaS.

| Pilot Project | Country/Region | Pilot Time | Travel Interaction Object | Ticket Service Type | MaaS Type |
|---|---|---|---|---|---|
| **Moovel** | Hamburg/Stuttgart (Germany) | 2015- | Car-sharing/taxi; Public transit; Train | One-way/month ticket | MaaS 2.0 |
| **MyCicero** | Italy | 2015- | Public transport; Domestic/international transport; | One-way/month ticket | MaaS 2.0 |
| **NaviGoGo** | Scotland (UK) | 2017- | Car-sharing/taxi; public transit; Train | One-way/month ticket | MaaS 2.0 |
| **IDPASS** | France | 2017- | Car rental/taxi/valet parking | One-way/month ticket | MaaS 2.0 |
| **SMILE** | Vienna (Austria) | 2017- | Ride-hailing/sharing bikes; Public transport in the city; parking | One-way ticket | MaaS 2.0 |
| **Whim** | Birmingham (UK) | 2018- | Car-sharing/taxi; Sharing-bike; Public transit | One-way/month ticket | MaaS 3.0 |
| | Helsinki (Finland) | 2016- | Car rental/taxi/sharing-bike; Public transit; Train | One-way/month ticket | MaaS 3.0 |
| | Antwerp (Belgium) | 2019- | Car-sharing/taxi/sharing-bike; public transit | One-way/month ticket | MaaS 3.0 |
| **Bus card** | Shanghai (China) | 2017- | Public transit | Daily/three-day Ticket | MaaS 2.0 |
| **Maishi** | Shenzhen (China) | 2018- | Public transit; Sharing-bike; Minibuses | One-way ticket | MaaS 2.0 |
| **Beijing MaaS** | Beijing (China) | 2019- | Public transit; Sharing-bike; Ride-sourcing; suburban railway | Just information inquiry | MaaS 1.0 |
| **Ctrip** | China | 2017- | Domestic/international mobility; Ride-sourcing | One-way ticket | MaaS2.0 |

*2.3. Foundations of Participators and Technologies in MaaS System*

In order to address the questions described above, we must first tackle the sticky problem of understanding how to treat MaaS as a concept that currently lacks a formal and robust definition. MaaS is often described as an alternative to private vehicle ownership that combines different types of mobility services as part of a single, seamless offering made available to users via subscription-based smartphone applications and is also referred to using the rubrics "combined" or "integrated" mobility services. In the past five years, MaaS has attracted significant attention as it holds the potential to disrupt the business model of the traditional automotive industry and the way people travel.

As shown in Figure 1, the ecological system of MaaS consists of several participators, including (1) mobility service providers (including public/private transport service providers), (2) data service providers (traffic network data, user trip data), (3) MaaS platform operators (divided into diverse subjects such as mobility service providers, other third-party actors in the MaaS system, regulatory agencies and independent third-party operators), (4) ICT or insurance service providers, (5) government regulatory agencies. As the MaaS ecosystem evolves, other actors could also be added, such as media or advertising firms, investors and so on, but for the purposes of this paper, we will focus only on the heart of actors that could enable the MaaS system.

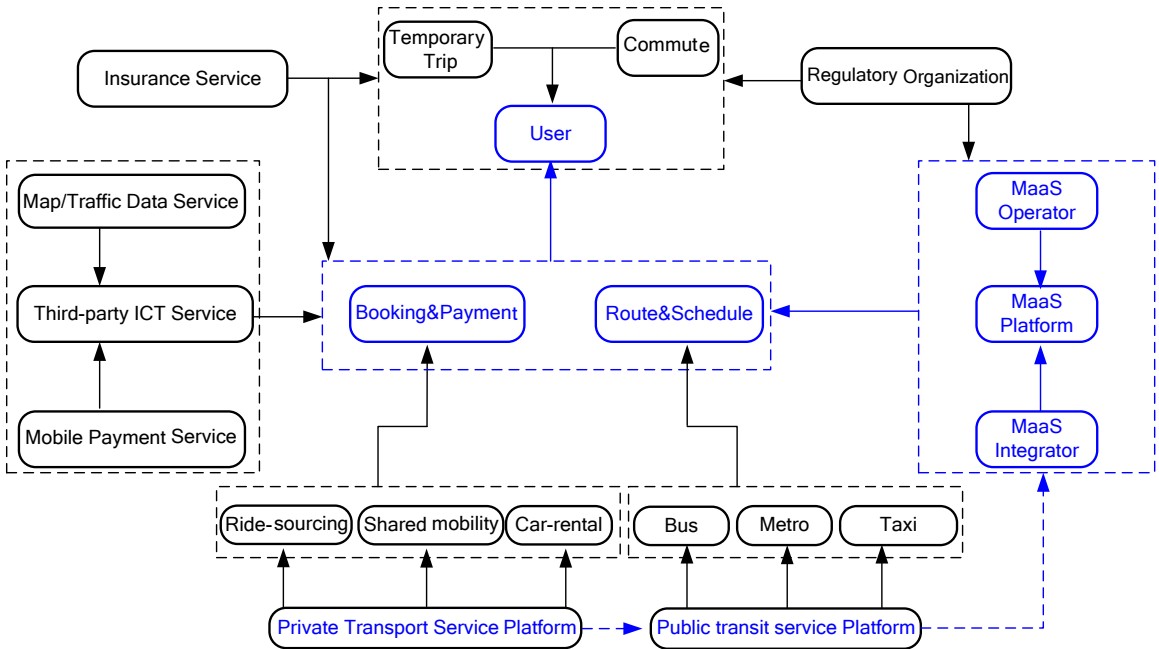

**Figure 1.** Actors in Ecological System of MaaS.

### 2.3.1. The Key Participators for MaaS System

Briefly, implementing and operating a MaaS system just requires a real-single identity for a user, open data and mobile payment from various mobility modes that hold interactive and collaborative principles. There are still some extra foundations that must be reached in order to implement and conduct MaaS successfully. The extra foundations to implementing a MaaS system in an urban area may be summarized as below: (1) large commuting demand and diverse mobility modes are available in the urban area; (2) the main mobility operators agree to share travel data including real-time data; (3) all mobility services in the MaaS system support mobile payment and identification of e-tickets. Considering the general conditions above, we conclude that the MaaS system, as an alternative choice to owning a car while enjoying equally convenient transport services, may only be operated in a city where commuters are willing to shift from car-dependent travel to public transit centric multi-mode travel. In other words, public transit is still the key component of the MaaS system, MaaS would only be conducted in those cities that own adequate public transit resources to allow potential users to travel more conveniently or at less cost than owning a car.

### 2.3.2. Three Key Technologies for MaaS Ecosystem

Prior to the emergence of the (MaaS) model, the travel field has formed a trend of cross-development of three major modes (mobility information system (MIS), sharing mobility mode (SMM) and mobile communication technology (MCT)) (see in Figure 2 and Table 3). Currently, the user has to use numerous tools of MIS, SMM and MCT in

order to find information and purchase and access different mobility modes. In order to satisfy the demand for diverse travel patterns and improve operation and management level and operational efficiency of transport infrastructure, MIS integrate the information between the urban-related travel of all kinds of mobility information and road congestion information to improve urban travel efficiency. Its main features include information queries for public transit schedule and other mobility supplies, real-time traffic condition, blocking traffic reporting and publishing, dynamic route guidance and navigation. MIS can provide more comprehensive, timely, convenient, interactive information services through the sharing of travel information. SMM is the most influential mobility choice in the world in recent years, it supports the sharing of vehicles with others refers to the shared ride way, in accordance with the mobility service fee requirements. It represents a large number of innovative modes containing a ride-sourcing service, shared bicycle/scooter service. On one hand, MIS meets the diverse demand of travelers; on the other hand, it helps increase effectively the use rates for idle vehicles to avoid the waste of resources. The development of MIT has provided a variety of auxiliary support for mobility choice behaviors, including the query of multi-modal traffic information; dynamic route guidance and navigation during driving; expanding functions in mobile payment patterns such as bus, taxi, parking and highway during traffic, which make travel more convenient and efficient. SMMs (such as ride-sourcing and car-sharing modes) are very promising business models for a more sustainable transportation system, it not only contributes to a more sustainable transportation system, but also can help to reduce the number of cars driving on the road, and improve traffic efficiency and passenger convenience.

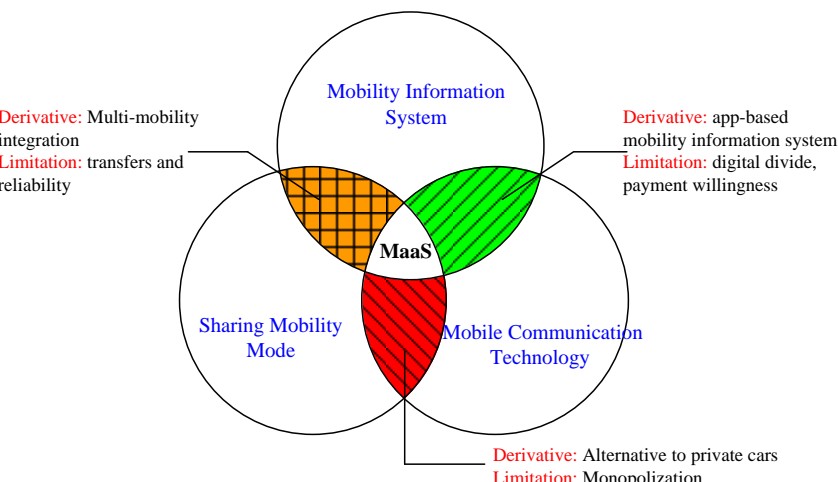

**Figure 2.** Three Emerging Technologies in MaaS Ecosystem.

　　Collaborative development of SMM and MCT forms many sharing-economic scenarios such as sharing-car, sharing-bike, sharing parking and other business models, leading to various operator platforms, payment methods, and reservation platforms. Due to the lack of integrity and uniformity, some operator problems such as bad competition, leakage of personal information and the coexistence and development between SMM and the traditional taxi modes make up the internal and external contradictions of SMM. At the same time, the issue of the digital divide phenomenon under these scenarios is also not resolved. The integration of MIS and SMM has promoted the transformation of travel mode selection from single-mode to multi-mode hybrids and has promoted the development of urban multi-mode transportation travel systems, but not the reliability of the connection between multi-mode travel modes (such as the last-mile problem, transfer inconvenience problem and difficulty in parking problem). By collecting static and dynamic service information of various travel modes, MIS and MCT provide current traffic and road conditions for travelers and help travelers optimize travel mode, travel time and route. In addition, the development of mobile payment technology simplifies the billing process of

travel services and improves travel efficiency. However, the willingness to pay and the digital divide of tourists reduce the popularity of the tourism information system.

**Table 3.** Existing and emerging technologies applications in the mobility area.

| Technology | Mode | Application | Application in China |
|---|---|---|---|
| MIS | Public transit information inquiry | Transit | Beijing bus |
| | Ticket payment | | Shanghai metro |
| SMM | Sharing bike | Limebike | Ofo; Mobike; Hellobike |
| | Sharing Scooter | Lime; Bird; Spin | - |
| | Ride-sourcing | Uber; Grabcar | Didi; Dida |
| | Carpooling | UberCommute | Didi-ridesharing |
| | Carsharing | Car2go | Gofun; EVCARD |
| | Microtransit | Moia; UberHOP; GrabShuttle; Lyft Shuttle | Didibus |
| | Automated Taxi | nuTonomy | - |
| MCT | Mobile payment | Paypal; Google checkout; Apple pay | Alipay; WeChat pay; Union pay |
| | Map and Navigation | Google map | Baidu map; Amap |

The emerging MaaS pattern integrates the advantages of MIS, SMM and MCT to break down barriers using optimal pricing schemes and then reduces travel costs, and enables users to purchase and get a convenient door to door service according to the travel required, thereby reducing the use of private vehicles, improving transportation resource utilization and ultimately achieving green and sustainable transportation wisdom. The emergence of this model represents the transformation of individuals from the high cost of ownership of travel tools to the consumption of travel behavior as a service. The MaaS platform is based on sustainable transportation concepts such as intelligent public transportation dispatch, personal habits analysis and green travel priority. It integrates the mobile payment capabilities to achieve travel itinerary planning, one-click planning of routes, seamless connection of public transportation and cost-effectiveness. Functions of MaaS such as key payment improve public travel satisfaction and improve the green travel experience.

## 3. Analysis and Comparison for Different Scenarios of MaaS

In order to take advantage of MIS, MMS and MCT technology breakthroughs, several new development scenarios have emerged recently. Almost all of the scenarios emphasize the important role of public transportation authorization in the implementation process of MaaS. As the manager and supervisor of the MaaS ecosystem, the public transportation authorization (PTA) will not only formulate relevant regulations in terms of the degree of supervision, management regulations, industry alliances, etc. it will also be involved in the development of the MaaS system and play an important role. The appropriateness of its role positioning plays a decisive role in the development of this novel travel mode. Meanwhile, there is an imperative need to design the MaaS ecosystem and identify the actors involved and their roles.

### 3.1. The Public/Private Ownership Scheme

Kamargianni et al. [45] introduced a MaaS provider entering into the MaaS ecosystem to balance the integrated relationship between the public transit service (PTS) and non-public transport service (NTS), he proposed PTS/NTS ownership patterns according to MaaS provider attributes. Based on the knowledge assumption that "public transit is still the most key component in MaaS system", the PTS ownership mode regards the public transport authorization (PTA) as the MaaS provider, which can not only guarantee that all

types of PTS in the urban area will be offered via the MaaS system but also it may be easier to secure the engagement of NTS (i.e., ride-sourcing; sharing-bike), due to the fact in most cities, PTA is the one responsible for procuring all the transport operators. Meanwhile, the PTS ownership pattern can also provide the potential opportunity for achieving the societal goals [44] of MaaS based on network efficiency and transport equity considerations. For example, as optimization measures of urban transport efficiency, peak charges (e.g., tolls in Singapore and London), road tolls (e.g., highways in China; high-occupancy toll lane in the US) and parking pricing have attracted researchers' attention in recent years, but there still has been limited implementation. Perhaps the greatest superiority under a PTS-based MaaS model is the collaborative ability of the PTA to reduce traffic congestion by incorporating road toll and parking pricing as an element in the package price, which makes the urban transport system (UTS) become more efficient and convenient. However, PTA may find it too difficult to diversify or extend their role under fair competition principles in the PTS pattern and it may slow down the innovation penetration in the process of the MaaS system. For example, the Swedish UbiGO pilot slowed its progress because of the intervention of PTA at the beginning of this project. In addition, the properties of non-profit organizations for the PTA make it probably not have the incentives to improve the services of the MaaS system. Another disadvantage of the PTS ownership pattern is the fact that the pattern transfer (expanding to other cities) is challenging to achieve. It is out of their scope to provide travel services that could be used in other cities as well. The independent MaaS platforms of each city cannot form a scale effect.

Compared to the PTS-based pattern, MaaS providers may be any one or more non-PTS operators that will diversify or extend their current travel services or may be third-party data or technology providers who want to provide travel services in the privatized NTS pattern (see in Figure 3b). The NTS pattern can accelerate the MaaS development process with its flexible and diverse property. In addition, it is easier for NTS providers to offer roaming services under driven-by-profit maximization. Clearly, accessing the support of PTA rather than improving service efficiency will become the key subject in the NTS ownership pattern of MaaS, which would hinder the development of MaaS. Otherwise, one greater risk is that PTA may go against the NTS pattern for fear of losing their reputation as the transport integrator and provider.

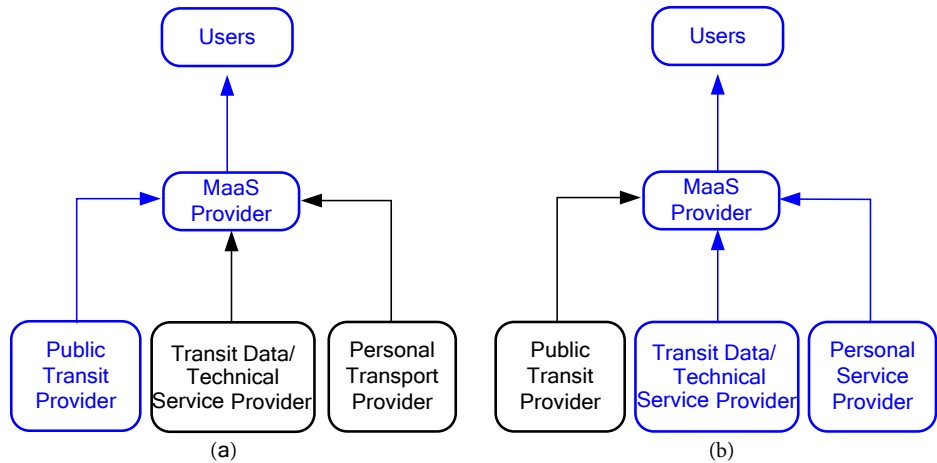

**Figure 3.** Kamargianni's Public/private Ownership Scheme: (**a**) PTS ownership pattern; (**b**) NTS ownership pattern.

## 3.2. The Government Engagement Scheme

Based on the research of Kamargianni et al. [45], Smith et al. [21] proposed a new business development scheme of MaaS, as shown in Figure 3, which they argue can facilitate discussions about MaaS, notably "comparisons of" different scenarios, as well as understanding the potential effects of MaaS. The MaaS integrator, introduced by Smith as

described to the board of directors, was to be organized as an intermediary marketplace that connects two types of actors, transport service providers (TSP) and MaaS operators. According to different levels of government engagement, in Smith's government engagement development pattern of MaaS, there are three different business development scenarios: public control pattern (PCP), market-driven pattern (MDP), public-private partnership pattern (PPP).

From the PCP of MaaS, we can recall traditional definitions in Kamargianni's PTS ownership pattern. They both generally agree that PTA at national, regional and local levels have key roles to play in potential transitions to the implementation of MaaS, regardless of their intended operative roles in the emerging MaaS system and public transit (PT) is still the main component of MaaS system. In this scenario, PTA would not only be responsible for planning and procuring traditional PTS, but also for adopting the MaaS integrator and MaaS operator roles (see Figure 3a). Furthermore, there will be more advantages in policy formulation and market regulation in the PTA. As shown in Figure 3b, MDP of MaaS implies that the MaaS integrator and MaaS operator roles are either absorbed by incumbent NPS providers, such as transport service providers or transit data/technological service providers in the PTS pattern of Kamargianni's MaaS system (see Figure 3), or attract new third-party organizations to be established for MaaS. In MDP, MaaS is being implemented in somewhat of a policy vacuum, driven by the market with limited authorities regulation. MaaS integrators and MaaS operators are motivated by a commercial imperative that may or may not align with social goals for optimization in transport efficiency and land use. Considering these defects, the third scenario, PPP, is interpreted as the intermediate pattern between the PCP and MDP scenarios. In the intermediate pattern of PPP, the PTA is regarded as the MaaS integrator and regulator that enlarges the scope of MaaS, while the MaaS operator role remains open for NTS providers (see Figure 4c). The potential benefit of this scenario that proponents foresee is that a publicly controlled MaaS integrator could act as a neutral buffer between private MaaS operators and transport service providers to mitigate the risks of MaaS operators in MDP.

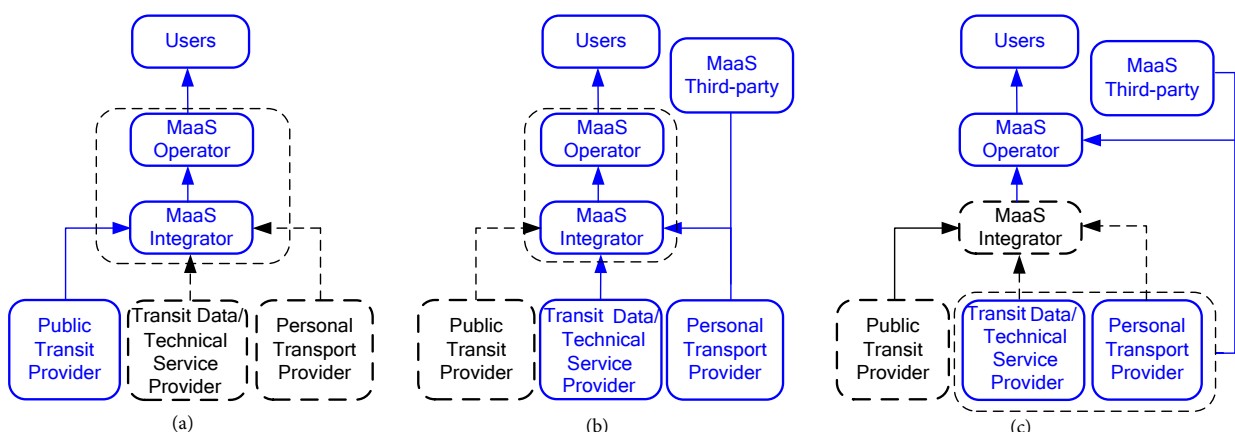

**Figure 4.** Smith's Government Engagement Scheme of MaaS: (**a**) public control pattern; (**b**) market-driven pattern; (**c**) public-private partnership pattern.

The logic of these three scenarios in Smith's MaaS development scheme mainly rests on three underpinning arguments. Firstly, the PT is always the key component of the transport system no matter if it is the traditional urban transport system or the emerging MaaS system. Secondly, the main purpose of the development of MaaS is to contribute to societal goals (green, sharing, humanism, sustainability) by facilitating a mode shift from private cars to travel services. Finally, PTS and NTS have different and conflicting goals. For instance, Smith et al. [21] explained that an MDP of MaaS might aim to maximize the revenue by selling as many travel packages as possible. In contrast, the PCP of MaaS rather strives towards reducing the travel amount of private cars and increasing the modal

share of PT, which is an inexpensive product compared to ride-sourcing or car-sharing. Moreover, some contend that the business opportunities in adopting the new roles in the MaaS ecosystem are limited or non-existent, due to small margins within the sector, large administration costs and a lack of proof of the end-users' willingness to pay [47]. Hence, the PCP of MaaS may be the better choice for the current development situation considering public funding subsidy might be needed to catalyze the development and diffusion of MaaS [21].

### 3.3. The Alliance-Based Scheme

This part can be used to answers the questions "Which business development paradigms (BDP) in China is better for different MaaS foundations?" and "Who should be the provider, integrator and operator of MaaS ecosystem in China?"

Firstly, we will answer the question "Which business development paradigms (BDP) in China is better for different MaaS foundations?" The two kinds of MaaS schemes described above by Kamargianni and Smith both regard the same major question concerning candidate actors (integrators or operators) in the role of government in the emerging MaaS system. Further, the barriers to inter-organizational cooperation between the public and private actors are particularly challenging due to the inherent differences, notably in terms of legal frame, bureaucracy and political control. Whilst some researchers advocate for the PTA to assume the broker role (refer to three Bs model bundles, budgets and brokers) [48], others have found by a Delphi study of experts that transport operators are the preferred service integrator, followed by a third-party travel provider [23]. While Smith et al. [47] acknowledge that the PTA should have an important role to play in encouraging innovation in the market rather than a primary role involved in the innovation process, there is strong support from the private sector to rely on private equity with appropriate regulations in place to encourage private sector activity.

Secondly, this paragraph answers the question "Who should be the provider, integrator and operator of the MaaS ecosystem in China?" Based on the above considerations, this paper proposes an alliance-based scheme of the MaaS system from the perspective of brands (see in Table 3), holding for existing travel providers and ensuring the government's functional attributes as shown in Figure 5. In contrast to the other MaaS schemes, the alliance-based scheme introduced in this paper is a combination constituted by several actors (travel service providers, transit data/technical service providers and other third-party organizations) with corporate goals taking profits and social benefits into account. They decide voluntarily to cooperate and share knowledge, assets and risks, accepting relatively more uncertainty regarding context and added value in the alliance of MaaS. Obviously, choices of potential actors with respect to the alliance are made within a strategic decision-making framework [28], not all potential actors of MaaS agree to join the alliance after strategic assessment. There are two patterns in our alliance-based MaaS scheme: independent operated pattern (IOP) and alliance operated pattern (AOP). In an IOP, any participators of the MaaS alliance can apply for authorized operation qualification of the MaaS system, and accept the supervision of public transport authorities (PTA). Paraphrasing, the existing travel service providers (including PTP and NTP) are regarded as the brokers for third-party travel services (TMSs) when they access authorized operation licenses for MaaS through the alliance, they can not only integrate their own travel products while retaining the premise of self-branding and self-platform but also attract TMS into their own MaaS platform, so as to sell all kinds of MaaS packages (see in Figure 5a). This pattern has opened up the key foundations of inclusive concept, policy formulation and data sharing to promote the transformation of all traditional travel service operators into emerging the MaaS scheme. It can be predicted that the rapid business development and large-scale expansion will make this innovative traffic mode more easily accepted by users. The shortcoming of this pattern may be that the IOP platforms might be unwilling to integrate those homogeneous travel services that compete with them, which will reduce the

choice diversity for users. In addition, the transformation efforts of small- or medium-sized travel service providers are also limited by their business capabilities.

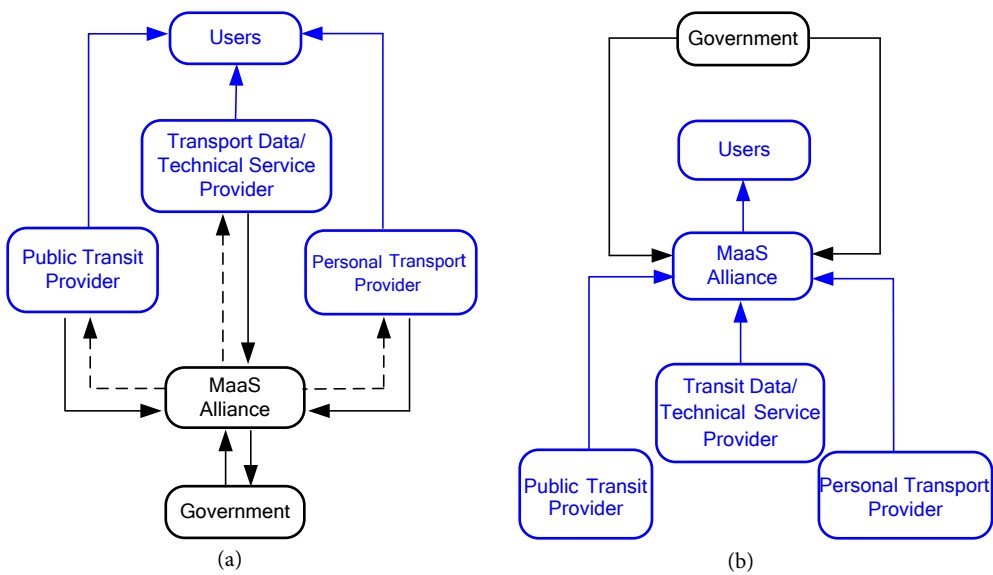

**Figure 5.** The Alliance-Based Scheme: (**a**) Self-owned brand operation pattern; (**b**) Alliance brand operation pattern.

Compared with the IOP, travel service providers or other participators in the alliance cannot obtain the authorized operation license of the MaaS platform in AOP. As the unique MaaS operator, The MaaS alliance will create a new third-party brand and platform to conduct mobility business. In this pattern, the MaaS alliance integrates all types of travel services through an open platform interface for internal union travel and all brands integrate services, which sell travel packages and services to users. Unlike the IOP, the homogeneous mobility services provided by different actors are differentiated in AOP, and the result of choices will be decided by end-users. The MaaS platform operated by the alliance acts as an intermediary in this pattern, it distributes the package fees paid by users to related travel service providers, and the role of the PTA in this model is the same as the IOP (see Figure 5b). The AOP of the MaaS system provides greater convenience and a better customer experience, but it also brings potential risks to the MaaS platform when third-party services are not provided as promised. The MaaS service model provided by the Finnish app Whim basically conforms to the AOP.

The MaaS scheme proposed in this section focuses more on the way of guiding functions in the government management attributes, rather than the compulsory driving role considered in the schemes of Section 4.1 or Section 4.2. At the same time, it takes into account the development of existing travel service brands, which is more conducive to the integration and operation of the MaaS mode. Public–private collaboration can be realized in the alliance-based scheme, which is helpful to effectively govern the development and diffusion of sustainable MaaS.

## 4. MaaS Pilot Experiences

To date, Sweden and Finland have acted as pioneers in the development of the MaaS system. For instance, the 2013 pilot of UbiGo in Gothenburg is often referred to as the first demonstration in real-life conditions for MaaS while in 2016, the launch of Whim in Helsinki drew international attention to the MaaS concept.

### 4.1. Sweden MaaS Pilot Project–UbiGo

In 2013, Sweden officially launched a one-stop travel service in Gothenburg, UbiGo, the first to introduce the monthly rental model of the telecommunications industry into the travel service industry, which provides users with door-to-door seamless travel services.

During the two-year project pilot process, 70 questionnaires, travel diaries and group interviews were conducted with 70 groups of local families in Gothenburg. The ideal test results were obtained and showed that nearly 50% of the families changed their original travel methods, 40% of households changed their mobility plans and 25% of households changed their travel chain; many participants gradually reduced their dependence on private cars, shifted more to PT, walking and green-cycling mobility modes. The pity is that UbiGo was not extended after the pilot period as the PTA had to determine what it was legally able, and strategically willing, to do.

### 4.2. Finland MaaS Global Project–APP Whim

People living in the Helsinki metropolitan area of Finland face the high cost of cross-regional travel. At the same time, too many private cars have also led to the problem of traffic jams in this area, which has seriously affected the normal life of residents. In 2016, the Helsinki Local Government (PTA) partnered with the Finnish travel services company MaaS Global to create the world's first MaaS platform app, Whim. Whim provides an open interface to access the platform through taxi/car rental companies, city bus company, shared cycling services, etc. In Whim, all PT and private shared travel services are bound into a monthly package, the user can travel by using the Whim service in downtown Helsinki. Convenient operation and humanized multi-mode selection make the utilization rate of the app continuously improve. In addition, WHIM also provides the corresponding package content and pricing scheme for different travel needs (see Table 3). Users can book one-way or monthly travel plans by binding a credit card or using Whim, and then pay for the trip at the end of the month, eliminating the need to lock in the travel fee in advance. The Whim platform's good operation made the Finnish government decide to sharply cut the use of private cars by 2025, thus creating a sustainable urban transportation network.

### 4.3. The Keys to the Success of MaaS Projects in Sweden and Finland

This part answers the question "How to learn from success and failure in the current pilot?" In a sense, MaaS is an intrinsically new business ecosystem based on open innovation that requires collaboration and support by multi-actor in order to bring MaaS offerings to end-users. The success of the MaaS projects in Sweden and Finland also depends on several factors, first of all, government encouragement and policy support. All actors in the MaaS system are supposedly motivated by MaaS's potential contribution to their organizational goals, whether these goals are reduced private car traffic (PTP), improved public-transport efficiency (PTA) or increased profit (NTP). The Finnish government issued traffic service data sharing regulations during the implementation of the MaaS project, forcing all transportation service providers to open their data and formulating management policies such as financial support, legal adjustments, support demonstrations, and deregulation; the PTA of Gothenburg also has a future traffic plan for sustainable urban mobility and a series of encouraging policies have been issued, which have prepared the ground for the implementation of UbiGo.

The second is the alliance cooperation among stakeholder actors in the MaaS system. The Swedish UbiGo pilot is part of the Swedish "Go Smart" project alliance, this alliance is a series of organizations composed of Lindholm Science Park, the Swedish Joint Innovation Agency and other related industries, academia and the public sector. Finland's Whim platform relying on the European MaaS Union earned more abundant financial support from the MaaS Development Fund, which was jointly set up by the Finnish innovation financing institutions, the Ministry of transport and communications of Finland and the Finnish Department of transportation. These advantages attract all kinds of traffic, encouraging Helsinki travel service providers to join and expand the alliance leading to the rapid advance of the MaaS Project progress.

Finally, there is a sound data sharing mechanism. The Swedish UbiGo and Finnish Whim platforms analyze the user's travel behavior based on the user's massive daily travel data and rationalize customized travel package services. It can better meet the different

needs of users to enhance users' willingness to pay. Conversely, users' travel chain data change has also helped travel service providers to improve the quality of service. Table 4 show us humanized tariff products (e.g., MaaS mobility package). In addition, from the Sweden Ubigo pilot experience, the effective way to promote the active participation of the public in the MaaS pattern is not only the satisfaction of travel services or the innovation of the mode, it also includes providing subsidies in PT fees. However, based on the social welfare attribute of PT services, The discussion on whether the commercialized MaaS pattern can also enjoy the financial subsidy has not been resolved. This also led to the UbiGo pilot project's failure to continue.

**Table 4.** App Whim pilot city travel package content and charging standards.

| Travel Services. | Helsinki (Finland) | | | Birmingham (UK) | | Antwerp (Belgium) | |
|---|---|---|---|---|---|---|---|
| | € 59/package | € 249/package | € 499/package | £ 99/package | £ 349/package | € 0/package | € 55/package |
| City bus | Unlimited | Unlimited | Unlimited | Unlimited | Unlimited | charge | Unlimited |
| taxi | 10 €/time/5 km | −15% discount | 80 times | No include | Unlimited | Charge | 10 €/time/5 km |
| Car rental | 49 €/day | Free weekend | Unlimited | £ 49/day | Unlimited | No include | 49 €/day |
| Bike sharing | Unlimited | Unlimited | Unlimited | Coming soon | Coming soon | Charge | 30 min/day |
| **Additional items** | | | | | | | |
| Vehicle booking | ✓ | | ✓ | - | - | - | - |
| Subscribe / Cancel | ✓ | ✓ | ✓ | ✓ | ✓ | ✓ | ✓ |
| Paid item | Shared scooter | Shared scooter | Shared scooter | - | - | Intercity train | Intercity train |

## 5. MaaS in China

As the world's largest mobility market, mobile app-based ride-sharing services have flourished in China relative to those in most countries (see Figure 5a) [49]. MaaS is not only potentially a massive opportunity for auto manufacturers and technology firms in China, but it also will change the Chinese transportation system dramatically. As shown below in Figure 5b, according to ARK's research model, Chinese MaaS will deliver USD 2.5 trillion in revenue by 2030, for perspective, global original equipment manufacturer (OEM) revenues are just USD 2 trillion today [49]. However, for now, China's MaaS projects are still in the discussion stage. In October 2018, the first MaaS platform in China, "Maishi" app beta, was launched in Shenzhen. It integrated conventional PT, customized PT, dynamic minibus, sharing-bike and other transport modes to provide personalized customized travel services for users, and also can be regarded as a market-driven pattern under Smith's MaaS development scheme because the private enterprises are the operation main body. However, the Maishi app is just in the internal trial stage right now and has not been implemented in reality due to concerns about the impact of such operations. On 4 November 2019, the Beijing transportation commission (BTC) and Alibaba subsidiary, amap company, jointly launched the first pilot MaaS application: an integrated green travel service platform in Beijing (MaaS-Beijing). It also can be regarded as Smith's public–private cooperation pattern or a simplified version of the self-owned brand operation pattern provided in this paper due to the fact that the MaaS products are integrated into the existing map platform and the operation main body and brand of the MaaS platform is an existing travel service provider. It should be noted that MaaS-Beijing is just in the MaaS 1.0 stage referring to the classification framework of MaaS [44], because it just provides the

service of information query for the bus schedule and subway congestion, not including ticket payment or long-time package.

### 5.1. The Opportunities and Foundation of MaaS in China

At present, the development of MaaS in China is facing a new revolution with the rapid development in road traffic, popularization of automobiles and coming of the new technical revolution and information era. This part also can answers the question "How to ensure that the city has a good implementing foundation for the MaaS ecosystem in China?"

In recent years, the Chinese government has launched a series of policy supporting documents to promote innovation and development in the field of travel vigorously and made efforts to become a powerful country in transport (see Figure 6). In September 2019, the Central Committee of the Communist Party of China and the State Council issued the Outline for the Construction of a Powerful Transportation Country (the "outline"), which aims at striving to develop shared transportation, building a service system based on mobile intelligent terminal technology and realizing mobility as a service [50]. The National Development and Reform Commission together with seven other government authorities published the Guiding Opinions on Promoting the Development of the Sharing Economy (the "Opinions") on 3 July 2017, Chinese government will take a series of encouraging and supportive measures to facilitate the development of the mobility area [51]. China's State Council released the Development Plan for Modern Comprehensive Transportation System during the 13th Five-Year Plan (the "plan"). The "plan" lists integrated, interconnected and efficient construction of comprehensive transportation as one of the important tasks of China's transportation development. By 2020, China will build a safe, convenient, efficient and green modern comprehensive transportation system [52].

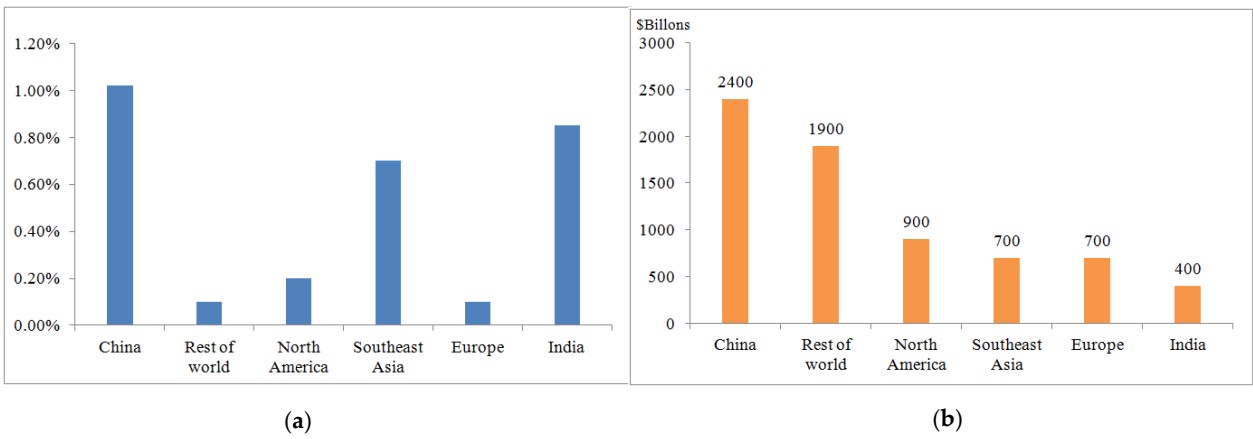

(**a**)  (**b**)

**Figure 6.** Current and Future Market of MaaS in China: (**a**) Proportion of MaaS Miles in Total Vehicle Miles Traveled in 2017 (**b**) Forecast Revenue of MaaS in 2030.

The development of MIS, SMM and MCT (introduced in Section 3.1) lay the foundations for the implementation of MaaS in China. As shown in Figure 7, in terms of the scale of mobile internet users, China's mobile phone users overview breaks 8.17 billion and accounted for 98.6% in 2018. Compared with 68% of the global average, the development of China's mobile internet is at the forefront of the world. Under the background of mobile Internet technology, China's mobile consumption has made rapid progress. In 2018, the scale of mobile payment reached RMB 208 trillion with a year-on-year increase of 45.2%. On the basis of the development of the mobile payment system, the development of new technologies has also brought rapid innovations in the entire mobility area, resulting in various types of transportation services such as sharing-bike, ride-sourcing, app-based taxis and customized buses. Judging from the market scale of mobile mobility and shared mobility, the number of mobile mobility users in China reached 216 million in 2018, and the travel service times relay on app-based taxis was 330 million, an increase of 43.37 million

compared with the end of 2017, a growth rate of 15.1%. The service times of ride-sourcing users reached 333 million, with a growth rate of 40.9%.

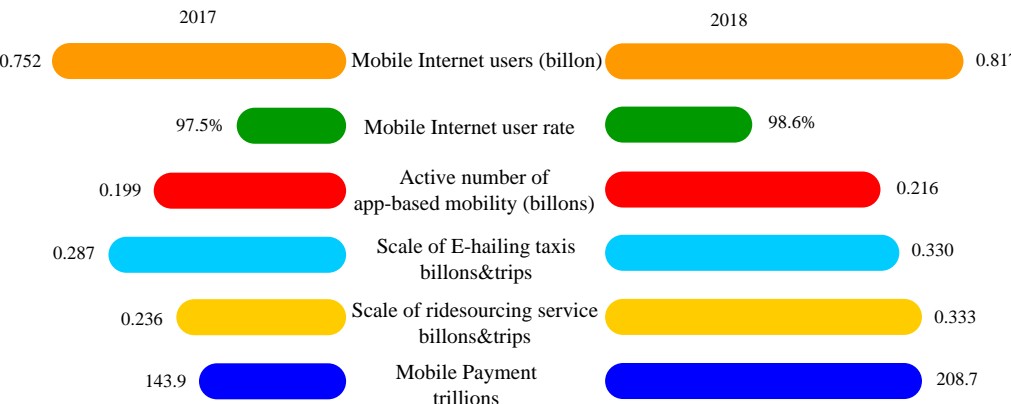

**Figure 7.** Status of scale development of China's mobile Internet, mobile travel, and mobile payment fields.

### 5.2. Corresponding Issues of MaaS Implementation in China

The successful pilot of MaaS in Finland and Sweden shows that the key issues including (but not limited to) seamless collaboration, sharing of data resources, business schemes selection, coordination, stakeholder development assessment and fully mechanized policy in security among participators of MaaS should be taken into consideration in the implementation process. On the other hand, this is also a key condition for the implementation of MaaS. The following are the issues that should be paid attention to when MaaS is implemented in China.

#### 5.2.1. Establishing a Long-Term Development Alliance and Coordinating the Interests of All Parties Demands

Referring to the European Union for Quality Industry and the Finnish and Swedish Union model, we can know that the coordination of interests of all stakeholders is the premise for promoting the implementation of the MaaS model. Among them, the government plays an important role in the alliance, including setting industry standards, defining entry barriers, process supervision, etc. The transportation service provider can maximize the social benefit while taking into account the social value. The public under various tourism services can get feedback of satisfaction with tourism services, and jointly form a healthy, green and sustainable transportation ecosystem.

#### 5.2.2. Evaluating the Prerequisites for MaaS Implementation and Determining the Development Path Model

This part can be used to answer the questions "How to achieve technology integration in China, such as driverless cars, block chain, artificial intelligence, etc?" and "Which business development paradigm (BDP) is better for different MaaS foundations?"

Although scholars and transportation researchers generally agree that the MaaS model can make future cities greener, smoother and more sustainable, the implementation of the MaaS project still faces major challenges in terms of operating conditions, user acceptance, development path and policy support. The data of operators in each travel mode is open, especially the real-time data sharing; travel mode operators allow third parties to sell their services; popularize coordination and communication in electronic ticketing and electronic payment. In order to get users to accept MaaS as a new travel pattern, not only does it need to be designed to meet the reasonable travel package demand, but also take into account the convenience of the MaaS interface based on foreign experience and network development rules about car models and it also needs to develop the necessary subsidy scheme. According to different operating conditions and operating conditions, it is

particularly important to set up a development path that conforms to the actual situation. This paper compares the mature experience of foreign countries and proposes a self-operated brand/alliance brand operation model, which is a development path that is more in line with the domestic situation. Before implementing the MaaS supporting policies, a comprehensive and scientific assessment of the implementation of the policy plan is required to reduce the potential social risks as much as possible and improve the reliability and credibility of the policy implementation. Drawing on the methods of the public policy field, there is a need to formulate a qualitative and quantitative pre evaluated framework and methods for policy implementation, and provide recommendations for management and operational decisions. Firstly, according to the demands of MaaS stakeholders, there is a need to formulate principles, standards and strategies for the sustainable development of MaaS. Secondly, the optimization policies need to be used to optimize the incentive policies. Finally, it proposes management countermeasures from the perspective of government industry management and urban operation.

### 5.2.3. Promoting Resource Development and Integration and Realizing Data Co-Construction and Sharing

This part can use to answer the question "How to advance data sharing for achieving the value of mobility big data in China?"

At the current stage, the lack of openness of the domestic transportation service industry, transportation management departments, and transportation operation data has not yet met the MaaS implementation conditions. In order to focus on exploring sustainable data utilization models, whether to integrate various data resources in the form of administrative orders, and how to reuse the data for the public, the development path is still being explored. From the experience of the Finnish Whim, China can achieve data disclosure by establishing a third-party operating platform and a data cloud platform. The holder of data information can be the transportation big data management organization or the basic service provider of the cloud platform.

### 5.3. Future Unanticipated Implications of MaaS in China

MaaS rhetoric promises unfettered freedom and instantaneous mobility to individuals within the context of a finite transport network. For example, MaaS Global advertises its application as "mobility on a whim", promoting directly this idea of individual unfettered freedom. However, this promise of freedom is on a collision trajectory with a challenge of simultaneous demand for travel in a transport network with a finite capacity, in which the main transport policy objectives are to reduce congestion (and the impacts that it has on emissions, journey times and urban quality of life) and to reduce GHG emissions and air pollution.

If efficiency is considered from the point of view of reducing network congestion and increasing vehicle usage rate, in the case that this reduces congestion and emissions by encouraging more use of non-car or shared vehicle modes through facilitating access then this could be positive for sustainability outcomes, that is, reducing congestion through making more efficient use of the existing vehicle fleet and increasing vehicle occupancy. However, this is contingent, thus some actors do temper their rhetorical support for MaaS as a business opportunity on the basis of data from other disruptive transport technologies.

The environmental consequences of any impacts on public transport use also need to be considered. There is evidence that public transport use is being reduced by the business models of ride-hailing brokers like Uber or Didi that make impulsive door-to-door service exceptionally convenient. Having shared vehicles available through MaaS could magnify this effect, leaving high-capacity, fixed-route modes in the cold. This would make the widespread adoption of shared self-driving vehicles much easier without necessarily reducing congestion. The spatial effects of this technological development are also in need of thorough investigation to ensure that further sprawl does not result.

## 6. Conclusions

This article analyzes the core foundation and development path of the MaaS system and introduces the Kamargianni model and Smith model based on the role transition of government functions. Considering the limitations of the models above, a new alliance-based framework of development scenarios in the MaaS system was proposed, which not only retains the government management and supervision functions of the original research but also makes full use of the existing travel service brand resources. It cannot be ignored that the most important things for potential stakeholders involved in an alliance-based scheme are cooperation and open data. In view of the fact that although the foundation of development services is there, a real MaaS practice has still not started in China, this paper compared the experience and lessons between the UbiGo project in Sweden and Whim project in Finland. By comparative study, we found that the key point to the success of MaaS projects is the cooperation of alliance, support of government policy and data sharing mechanism if the proposed alliance-based model is considered for the future Chinese MaaS scheme. It is undeniable that the MaaS is a complex ecosystem, and there will be many new problems in the process of operation, which need to coordinate the interests of all parties under the open and inclusive MaaS principle. Therefore, in order to obtain the goal of making service a part of sustainable transportation, the first priority is government intervention. Establishing an effective alliance-based MaaS strategy under government intervention and then conducting an economic analysis will become the focus of future research. In addition, we also noticed the role of the MaaS model in promoting the development of autonomous driving technologies (Singapore Autonomous Driving Pilot), closing the digital divide and promoting the transformation of traditional car manufacturers in MaaS service providers (Toyota's transformation). These positive signals undoubtedly make us look forward to the future of mobility as a service model.

**Author Contributions:** Conceptualization, Z.Z. and N.Z.; methodology, Z.Z.; validation, Z.Z. and N.Z.; formal analysis, Z.Z.; investigation, N.Z.; resources, Z.Z.; data curation, Z.Z.; writing—original draft preparation, Z.Z.; writing—review and editing, Z.Z. and N.Z.; visualization, Z.Z.; supervision, N.Z.; All authors have read and agreed to the published version of the manuscript.

**Funding:** This research was funded by National Natural Science Foundation of China, grant number 70971003.

**Data Availability Statement:** The data of China's mobile Internet, mobile travel, and mobile payment fields are collected from "Statistical Report on the development of Internet in China" (http://cnnic.cn/gywm/xwzx/rdxw/20172017_7056/201902/W020190228474508417254.pdf accessed on 8 April 2021).

**Acknowledgments:** The authors want to thank the Major Program of the National Natural Science Foundation of China (70971003).

**Conflicts of Interest:** The authors declare no conflict of interest.

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
