# Peer review of "A Novel Development Scheme of Mobility as a Service: Can It Provide a Sustainable Environment for China?"

_sustainability, doi:10.3390/su13084233_

Round 1

Reviewer 1 Report

The paper reads well and major part of the paper focus on discussing MaaS system in various aspects such as its components, operational framework and various stakeholders, government role and how it could facilitate successful MaaS operations. The paper also provide context of China, and discusses MaaS implementation requirements within China. There are a few concerns which are given below:

1) The research questions formulated in the introduction section seems very important and for a single paper they seems to be too many. Each questions in my opinon demands to analyze in a more rigorous approach. The paper tried to address each question and therefore significantly lacks details and rigor on each research question. It seems that authors summarizes the existing literature to discuss each question, and it is quite limited in a sense that if offer something original.   

2) Well, authors claim to present an improved version of MaaS Ownership and government engagement scheme, which also seems to be learned from some pilot schemes in Finland discussed in the paper. However, It would be more appropriate to provide more analysis using cases from other regions to strongly support authors claims and rigor of thier proposed framework. I think this seems to be the only contribution of the paper so It would be better if authors also elaborate section 4.3.

3) Section 3 seems redundant in my opinion, the information provided can be concised and should be made part of section 2. 

4) Please elaborate the sentence start at line No. 39 especially what is meant by incosistency here,.. "The lack of integration and inconsistency in the conncetion process between transportation modes..."

5) There are several typos, problem in referencing and spelling mistakes. Paper require a thorough revision in that aspect. A few examples are given below:

  • Line 91, different needs to be replaced with "difficult".
  • typo in line no. 128
  • Spelling mistake for word "integrate" line no. 158
  • Refernce need to be properly cited (please see line No. 306)
  • Tables No. need correction.

Author Response

Dear Reviewer

I have already revised my paper according to your Comments and Suggestions.

For comment (1)

 I've rearranged every question and make them closer to China's reality.

  • How to ensure that the city has a good implementing foundation ofMaas ecosystem in China ?  
  • Who should be the provider, integrator and operator of MaaS ecosystem in China?
  • Which business development paradigms (BDP)is better for Chinese MaaS foundations?  
  • How to learn from success and failure in the current pilot?
  • How to achieve technology integrationin China, such as driverless, block chain, artificial intelligence, etc ?
  • How to advance data sharing for achieving the value of mobility big data in China?

 In the Section 5.1, we answers the question”How to ensure that the city has a good implementing foundation for Maas ecosystem in china? “

In the Section5.2.2, we answer the questions”How to achieve technology integration, such as driverless,block chain, artificial intelligence, etc in china?”and“Which business development paradigms (BDP)is better for different MaaS foundations”.

In the Section5.2.3, we answer the question “How to advance data sharing for achieving the value of mobility big data”

In the Section4.3, we answers the questions “who should be the provider, integrator and operator of MaaS ecosystem in China?”and “Which business development paradigms (BDP) in China is better for different MaaS foundations”.

For comment (2)

we also elaborate section 4.3

For comment (3)

Section 3 are made as part of section 2

For comment (4)

we elaborate the sentence line 39

For comment (5)

We submit a revision to fix several typos, problem in referencing and spelling mistakes

Reviewer 2 Report

This paper presents an interesting overview of MaaS implementation in the world and its potential of implementation in China. Some minor changes are needed from my perspective.

Abstract is too long from my point of view. Some sentences can be the part of Introduction, while Abstract should point out the aim of the paper.

Technically paper needs to be cleared.

Keywords must be short...

Some sentences are too long, which make a reader hard to follow the authors idea (row 63-71).

Authors mentioned Kamargianni model and Smith model. It would be good to show their limitations and posibillities for implementation in China.

The most important factor in MaaS implementation can be government regulations. Can you please describe this factor on example of China. 

Authors stated few questions at the end of chapter Introduction. Are all answered for the case of China?

Author Response

Dear Reviewer

I have already revised my paper according to your Comments and Suggestions.

For comment (1)

I've rearranged the Abstract and keyword.

For comment (2)

 The long sentences(not limited to row 63-71) have been fixed.

For comment (3)

We discuss limitations of  Kamargianni model In section 3.1 (line298-310) and Smith model in section 3.2 (line350-354),otherwise We  also  analyzed why these two model are not suit for implementation in China in section 3.3(line384-390)

For comment (4)

 I've rearranged every question and make them closer to China's reality.

  • How to ensure that the city has a good implementing foundation ofMaas ecosystemin China ?  
  • Who should be the provider, integrator and operator of MaaS ecosystemin China?
  • Which business development paradigms (BDP)is better for Chinese MaaS foundations?  
  • How to learn from success and failure in the current pilot?
  • How to achieve technology integrationin China, such as driverless, block chain, artificial intelligence, etc ?
  • How to advance data sharing for achieving the value of mobility big datain China?

 In the Section 5.1, we answers the question”How to ensure that the city has a good implementing foundation for Maas ecosystem in china? “

In the Section5.2.2, we answer the questions”How to achieve technology integration, such as driverless,block chain, artificial intelligence, etc in china?”and“Which business development paradigms (BDP)is better for different MaaS foundations”.

In the Section5.2.3, we answer the question “How to advance data sharing for achieving the value of mobility big data”

In the Section 4.3, we answers the questions “who should be the provider, integrator and operator of MaaS ecosystem in China?”and “Which business development paradigms (BDP) in China is better for different MaaS foundations”.

Reviewer 3 Report

The issue of sustainable development plays an important role in socio-economic development research. Particular attention is paid to the transport systems, which, depending on the mode of transport, have a negative impact on the natural environment, reducing the quality of life, especially in urbanized areas. The authors tackled this issue by focusing on the potential of developing the Mobility as a Service (MaaS) in China. The chosen topic is an important one and still there are many gaps to be filled by the researchers in the area (one of the major ones being the lack of a unified definition of what MaaS is).

As the goal of the paper is not clearly stated, it is difficult to assess if it was fulfilled. Stating the goal, either in abstract or introduction would help to assess what one should expect from reading the reviewed paper. The sentence that resembles the goal from the perspective of the whole argumentation is in lines 649-652 – if this is indeed the intention of the authors it should be moved (or repeated) closer to the beginning of the paper.

As far for the authors' argumentation, they did fine work of reviewing the current state of the literature on the Mobility as a Service (MaaS) topic but said review is a bit too long and authors are prone to repetitions (e.g. section 2.2. about different pilot studies and then whole section 5. also dealing with particular pilot studies in more detail – are both needed?). By comparing two different models proposed by different authors (Kamargianni and Smith) authors proposed their own model of the alliance-based MaaS scheme. The presentation of two base models is not well executed, even with the provided diagrams the differences are not obvious – maybe including some direct comparison between all three models (as a table perhaps?) would made important differences stand out more and at the same time would allow for shortening the descriptions. The presented two case studies (Swedish and Finnish ones) serve as good examples of the finer points of the authors' proposes MaaS scheme. The connection between described cases and the situation considering the Chinese market could be strengthened a bit more (but that is a minor problem), as it is later explained that those pilot projects were presented as an example of the mature schemes in the face of China lacking its operational MaaS scheme.

It is possible to follow the authors' arguments, but it is not easy. Numerous spelling and language problems are the main drawbacks of the reviewed paper. Said problems range from fairly simple ones as missing commas, inappropriate use of capital letters (or lack of such use), repetitions (like using twice the term “operating conditions” in line 648), simple misspelling (like using the word “feathers” instead of “features” in line 133 and 146) or using words with similar meaning to those that should be used (like world “authorization” used instead of “authorities” in line 299 or “Travel” instead of “Mobility” used in line 301) to more serious ones like 5 to 8 line long sentences leaving the reader confused about what the authors meant (like 33-39, 64-70, 146-152, 250-254, 266-271, 475-497, 489-493, 581-585, 620-625, 625-629, 671-675). The thorough language verification is recommended, preferably done in cooperation with the authors and by someone at least acquainted with transportation/travel themes in order to assure that the quality of the argumentations would be upheld.

The conclusion is more like the summary of the article without providing insight about findings (those are presented in section six), potential stakeholders that may be interested in this study, limitations of this study, or further research avenues. Authors should try to address these issues, i.e. by clearly stating if the proposed model can be considered for the future Chinese MaaS scheme, or what conclusions would be important for potential stakeholders involved in this scheme.

The referenced sources are appropriate for the topic undertaken in the paper.

To sum up, the paper deals with the important and up-to-date issue and there is certainly an added value in that research: the proposed model and analysis of the potential for MaaS scheme on the Chinese market. That said, it should be noted that this paper would use some more work – especially by improving the presentation of the argumentation both by some reworks with the review part and improving the quality of the language.

Author Response

Dear Reviewer

I have already revised my paper according to your Comments and Suggestions.

For comment (1)

 I've rearranged the section 5.2.2 ”Evaluating the prerequisites for MaaS implementation and determining the development path model”and delete line 659-652

For comment (2)

We discuss limitations of  Kamargianni model In section 3.1 (line298-310) and Smith model in section 3.2 (line350-354),otherwise We  also  analyzed why these two model are not suit for implementation in China in section 3.3(line384-390)

For comment (3)

We submit a revision to fix several typos, problem in referencing and spelling mistakes( Not limited to line 33-39, 64-70, 146-152, 250-254, 266-271, 475-497, 489-493, 581-585, 620-625, 625-629, 671-675). and we also replace the inappropriate words in line 648, line 133, line146, line 299and in line 301

For comment (4)

We've reorganized the section of conclusion, and the main concerns of this paper are emphasized. The following is the main content of the conclusion:

“This article analyzes the core foundation and development path of the MaaS system, and introduces the Kamargianni model and Smith model based on the role transition of government functions. Considering the limitation of the models above, a new  alliance-based framework of development scenarios in MaaS system was proposed”

“It can't be ignored that the most important things for potential stakeholders involved in alliance-based scheme is cooperation and open data. In view of the fact that there is not  a real MaaS practice in China, this paper compared the experience and lessons between the UbiGo project in Sweden and Whim project in Finland. We found that the key point to the success of MaaS projects is the cooperation of alliance, support of government policy and data sharing mechanism if the proposed alliance-based model are considered for the future Chinese MaaS scheme. Authors should try to address these issues, i.e. by clearly stating if the proposed model can be considered for the future Chinese MaaS scheme, or what conclusions would be important for potential stakeholders involved in this scheme.”

Round 2

Reviewer 3 Report

The abstract was improved and it now better reflects the content of the paper.

The scope of the paper’s goal is better presented in the abstract, it is still very broadly shown in the introduction (as a set of questions). By highlighting which parts/sections of the paper provide the answer for particular questions, Authors have shown better connection between their argumentation/reasoning and the stated goals/questions to answer. However, maybe the broadening of the section titles to include said questions is a bit too much, the previous shorter versions of “The Alliance-based Scheme” and “The Keys to the Success of MaaS Projects in Sweden and Finland” would be recommended (minor thing, the number of section “The Alliance-based Scheme” is still 4.3).

Conclusion was improved, but it still is rather short and does not provide further research venues.

The Authors corrected some of the language issues, yet many are still left (mostly minor things, like using singular version instead of plural, incorrect use of capital letters and so on)  – there is a high chance they will be removed in the process of text editing and proofreading.

Author Response

Dear Reviewer

I have already revised my paper according to your Comments and Suggestions.

(1)the sub-title of section 3.3 and3.4 has been changed to“The Alliance-based Scheme” and “The Keys to the Success of MaaS Projects in Sweden and Finland” . Otherwise,  the stated goals/questions provided in the introduction has been  answered  more clearly in the section 3.3 and 3.4 

(2)The title number of“The Alliance-based Scheme” has been change to3.3

(3)Conclusion has been expended,and the future research areas are highlighted

“ How to establish an effective alliance-based Maas strategy under government intervention and then making an economic analysis will be the will become the focus of future research”

(4)Once again, we corrected some language problems